# Patterns of Care and Outcomes of Intensity-Modulated Radiotherapy and 3D Conformal Radiotherapy for Early Stage Glottic Cancer: A National Cancer Database Analysis

**DOI:** 10.3390/cancers11121996

**Published:** 2019-12-12

**Authors:** Mark C. Korpics, W. Tyler Turchan, Michael K. Rooney, Matthew Koshy, Michael T. Spiotto

**Affiliations:** 1Department of Radiation and Cellular Oncology, University of Chicago Medical Center, Chicago, IL 60637, USA; mkorpics@radonc.uchicago.edu (M.C.K.); wturchan@radonc.uchicago.edu (W.T.T.); rooney10@uic.edu (M.K.R.); mkoshy@radonc.uchicago.edu (M.K.); 2Department of Radiation Oncology, University of Illinois Hospital and Health Sciences System, Chicago, IL 60612, USA

**Keywords:** radiotherapy, laryngeal neoplasms, carcinoma, squamous cell

## Abstract

Early stage glottic cancer has traditionally been treated with 3D conformal radiotherapy (3DCRT). However, intensity-modulated radiotherapy (IMRT) has been recently adopted as an alternative to decrease toxicity. Here, we compared the usage and outcomes of IMRT and 3DCRT for patients with early stage squamous cell carcinoma (SCC) of the glottic larynx. Using the National Cancer Database, we identified patients with Stage I–II SCC of the glottis who received 55–75 Gy using IMRT (*n* = 1623) or 3DCRT (*n* = 2696). Median follow up was 42 months with a 5-year overall survival (OS) of 72%. Using a nominal logistic regression, race, ethnicity, year of diagnosis and fraction size were associated with the receipt of IMRT (*p* < 0.05). Using Kaplan–Meier methods and Cox proportional hazards models as well as a propensity matched cohort, there was no difference in OS for patients who received IMRT versus 3DCRT (hazard ratio (HR), 1.08; 95% confidence interval (95% CI), 0.93–1.26; *p* = 0.302). However, there was a survival benefit for patients receiving slight hypofractionation as compared to conventional fractionation (HR, 0.78; 95% CI, 0.69–0.92; *p* = 0.003). IMRT was associated with similar survival as 3DCRT, supporting the implementation of this potentially less toxic modality without compromising survival.

## 1. Introduction

Laryngeal cancer has a global incidence of 238,150 cases per year with glottic cancer representing approximately one-half of all laryngeal cancers [1,2]. Early stage squamous cell carcinoma (SCC) of the glottis is often treated with radiotherapy (RT) alone [3] and has traditionally been treated with 3D conformal radiotherapy (3DCRT) with excellent rates of long-term local control [4,5]. Generally, patients with early stage glottic SCC have a low rate of acute and late treatment toxicities. Nevertheless, patients are at risk to develop severe long-term side-effects of dysphagia, laryngeal/soft tissue necrosis and carotid artery stenosis leading to cerebrovascular events [4,6,7,8,9,10,11]. Recently, intensity-modulated radiotherapy (IMRT) has been adopted as an alternative modality to decrease toxicity, including strategies such as reducing dose to the carotid arteries and radiating a single vocal cord [12,13,14,15]. The goal of IMRT in glottic cancer is to prevent long-term toxicity without compromising already excellent oncologic outcomes. However, the implementation and outcomes of IMRT for glottic cancers remains unresolved.

In this work we compare the usage and outcomes of patients with early stage glottic SCC treated with IMRT and 3DCRT in the United States (US) using the National Cancer Database (NCDB). Specifically, we evaluated patterns of care and the association of overall survival (OS) with each modality.

## 2. Results

### 2.1. Patient Cohort Characteristics

A total of 4319 patients with cTis-2, N0, M0 glottic SCC who were treated with external beam RT (IMRT or 3DCRT) from 2004 to 2016 met eligibility criteria; 1623 patients (38%) received IMRT and 2696 (62%) received 3DCRT. Clinical and demographic characteristics are provided in Table 1. Comparing patients who received IMRT with those who received 3DCRT, there was a statistically significant difference in age, race, ethnicity, location of residence, year of diagnosis, treatment facility type and fraction size (*p* < 0.05). Patients treated with IMRT had a median fraction size of 2.00 Gy (interquartile range (IQR), 2.00–2.25 Gy), while patients treated with 3DCRT had a median fraction size of 2.23 Gy (IQR, 2.00–2.25 Gy). The median total dose for patients receiving 2.25 Gy fractions was 63 Gy (IQR, 63–63 Gy), and the median total dose for patients receiving 2.00 Gy fractions was 68 Gy (IQR, 66–70 Gy).

### 2.2. Utilization of IMRT

From 2004 to 2009 the proportion of patients receiving IMRT as compared to 3DCRT increased. Out of all patients diagnosed in 2004, 14% received IMRT and 86% received 3DCRT as compared to patients diagnosed in 2009, in which 39% received IMRT and 61% received 3DCRT. After 2006, the average use of IMRT per year remained roughly stable with a mean of 39% of patients per year being treated with IMRT. Figure 1 depicts the utilization of IMRT and 3DCRT by year of diagnosis.

### 2.3. Factors Affecting Selection of Radiotherapy Modality

Univariable and multivariable logistic regression for factors associated with the receipt of IMRT are described in Table 2. On univariable analysis, the receipt of 3DCRT was associated with patients >70 years of age (*p* = 0.008), high school education (*p* = 0.018), Western United States (*p* = 0.006) and slightly hypofractionated regimens (*p* < 0.0001). Receipt of IMRT was associated with black race (*p* = 0.001), Hispanic ethnicity (*p* < 0.0001), later treatment eras (*p* < 0.0001) and increased distance from the treatment facility (*p* = 0.048). On multivariable analysis, receipt of 3DCRT was associated with treatment at comprehensive cancer centers (odds ratio (OR), 0.72; 95% confidence interval (95% CI), 0.55–0.95; *p* = 0.019) and slightly hypofractionated regimens (OR, 0.56; 95% CI, 0.48–0.65; *p* < 0.0001). Receipt of IMRT was associated with black race (OR, 1.32; 95% CI, 1.04–1.67; *p* = 0.023), Hispanic ethnicity (OR, 1.77; 95% CI, 1.24–2.52; *p* = 0.002) and later treatment eras (OR, 2.77; 95% CI, 2.17–3.55; *p* < 0.0001).

### 2.4. Survival Outcomes

The median follow-up for the entire cohort of 4319 patients was 42 months with a 5-year OS rate of 72%. The Kaplan–Meier OS curve is shown in Figure 2, which demonstrates no significant difference in OS based on the RT modality used (*p* = 0.251). When stratifying by the use of standard fractionation (200 cGy per fraction) versus slight hypofractionation (225 cGy per fraction), the 5-year OS rates were 70% and 76%, respectively (*p* = 0.002). Figure 3 shows the Kaplan–Meier OS curves stratified by RT fractionation. However, when only including patients treated with slight hypofractionation, there was no significant difference in OS between IMRT and 3DCRT (*p* = 0.342), as shown in Figure 4. Univariable and multivariable analyses for OS are depicted in Table 3. After controlling for age, gender, race, insurance status, income, population density, treatment facility type, Charlson–Deyo comorbidity score (CDCS), clinical stage group, clinical T-stage and RT fractionation, there was no significance difference in survival for patients receiving IMRT or 3DCRT (hazard ratio (HR), 1.08; 95% confidence interval (95% CI), 0.93–1.26; *p* = 0.302). On multivariable analysis, age >70 years of age (HR, 2.17; 95% CI, 1.49–3.15; *p* < 0.0001) and CDCS ≥2 (HR, 1.76; 95% CI, 1.35–2.29; *p* < 0.0001) were associated with worse OS, while private insurance (HR, 0.60; 95% CI, 0.37–1.00; *p* = 0.049), income ≥$63,000 per year (HR, 0.78; 95% CI, 0.62–0.97; *p* = 0.028) and slight hypofractionation (HR, 0.78; 95% CI, 0.69–0.92; *p* = 0.003) were associated with improved OS.

### 2.5. Propensity Score-Matched Analysis

After matching for all covariates in Table 1 with *p* < 0.1, there remained no statistically significant difference in OS for patients treated with IMRT versus 3DCRT (HR, 0.90; 95% CI, 0.79–1.03; *p* = 0.127). Table 4 depicts the propensity score-match groups with no statistically significant difference between the groups. There was a total of 1428 matched cases included in the analysis with 714 records in each group.

## 3. Discussion

We used the NCDB to evaluate patterns of care for patients with early stage glottic SCC. We also estimated the OS differences between IMRT and 3DCRT. On both multivariable survival analysis and propensity score-matched analysis, there was no difference in OS for patients treated with IMRT as compared to 3DCRT. The goal of IMRT is to reduce toxicity without compromising treatment outcomes, and our findings support that OS is not compromised with IMRT.

Previous work has demonstrated that IMRT improves toxicity outcomes in head and neck cancer by, for example, reducing xerostomia [16,17,18]. In the setting of early stage glottic SCC, IMRT has the potential to reduce dose to the carotid arteries, which would potentially decrease the risk of stroke and more easily allow for future RT in the event of a second head and neck cancer diagnosis [4,6,7,8,9,10,11]. Others have shown that clinical outcomes are not compromised with IMRT, and dosimetric analyses demonstrate significant dose reduction to the carotid arteries by up to 75% [4,13,19,20]. Mohamed et al. performed a retrospective case-control study comparing 3DCRT and IMRT, showing no statistically significant difference in oncologic outcomes (e.g., 3-year locoregional control and OS); however, there was also no statistically significant difference in the rate of cerebrovascular events (*p* = 0.7), feeding tube dependence (*p* = 0.5) or aspiration events (*p* = 0.4) [19]. Zumsteg et al. also performed a clinical comparison of IMRT and 3DCRT and found no difference in oncologic outcomes, with 3-year local control rates of 88% and 89%, respectively (*p* = 0.938), while maintaining low doses to the carotid arteries [13].

The goal of using IMRT for early stage glottic SCC is to reduce treatment-related toxicity. However, to our knowledge, there are no prospective data showing a decrease in toxicity for IMRT as compared to 3DCRT for patients with early stage glottic SCC. The NCDB does not provide data on toxicity from radiotherapy; thus we are unable to make conclusions on whether or not IMRT resulted in less toxicity as compared to 3DCRT. Despite lack of data demonstrating decreased toxicity with the use of IMRT, the use of IMRT has increased over recent years as shown in Figure 1. While some studies have utilized generous margins including the entire larynx plus 1 cm, new studies are investigating radiating a single vocal cord as opposed to the entire larynx [12,13]. Al-Mamgani et al. performed a retrospective analysis investigating the use of IMRT to a single, entire vocal cord for patients with T1a glottic cancer [12]. After a median follow-up of 30 months, the 2-year local control rate was 100%. There was no grade 3 toxicity, and there was grade 2 acute dermatitis or dysphagia in 17% of patients. These patients were compared to an older cohort of patients treated to the whole larynx with 3DCRT, though not all patients were treated with slight hypofractionation [21]. This rough comparison demonstrated comparable local control (*p* = 0.24) but more acute toxicity in the 3DCRT group (66% vs. 17%, *p* < 0.0001). 

There is a clear survival benefit for patients with early stage glottic SCC treated with slight hypofractionation as opposed to conventional fractionation [22]. Our findings support this claim, as shown in Figure 3. Interestingly, we found that patients treated with IMRT were less likely to be treated with slight hypofractionation (OR, 0.56; 95% CI, 0.48–0.65; *p* < 0.0001). These observations suggest providers are more likely to use conventional fractionation when employing IMRT. Despite this finding, there was no survival difference in OS between IMRT versus 3DCRT overall (Figure 2, *p* = 0.251) or among patients who received 225 cGy per fraction (Figure 4, *p* = 0.342). 

The current study is subject to the inherent limitations of a population-based retrospective study; however, it provides motivation to perform prospective studies comparing IMRT and 3DCRT for patients with early stage glottic SCC given the lack of clinical data. Furthermore, the NCDB dataset contains only survival data, which is a less informative endpoint to evaluate the oncologic results of radiotherapy techniques in patients with early-stage glottic cancers. Patient data for toxicity, local control and cancer-specific survival would better facilitate comparisons between IMRT and 3DCRT. Nevertheless, disease recurrence in early stage glottic cancers does impact survival [23], even if this impact is small. Consequently, the high patient numbers in the NCDB facilitate the detection of small differences in survival associated with different radiotherapy techniques. To this end, using the NCDB, we and others observed survival differences in patients treated with different fraction sizes which were not observed in the original randomized trials addressing this issue [22,24,25,26]. Although the impact of IMRT on local control and toxicity is unknown, we did not find any difference in OS between IMRT and 3DCRT. 

## 4. Materials and Methods 

### 4.1. Study Design and Patient Data Set

We used the NCDB to perform a retrospective cohort study. The Commission on Cancer (CoC) of the American College of Surgeons and the American Cancer Society created the NCDB as a clinical oncology database sourced from CoC-accredited hospital registry data, including approximately 70% of all new invasive cancer diagnoses in the US [27]. Collected data include demographics, comorbidities, tumor characteristics, staging details, surgical and adjuvant treatments (e.g., radiotherapy) and OS. At the time of analysis, patient data were available for cases diagnosed from 2004 through 2016. 

### 4.2. Patient Selection

Patients with cT0–2, N0, M0 glottic SCC who were treated with definitive RT alone (3DCRT or IMRT) to a dose of 55–75 Gy were included in the analysis. Patients were excluded if they did not receive their first course of treatment at the reporting facility, were treated with definitive surgery, received chemotherapy or received immunotherapy. Figure 5 depicts the Consolidated Standards of Reporting Trials (CONSORT) diagram for inclusion and exclusion criteria.

### 4.3. Exposure Variables of Interest

Using the National Cancer Database, we identified patients with Stage I–II SCC of the glottis who received 55 to 75 Gy using IMRT (*n* = 1254) or 3DCRT (*n* = 3043). Patient, tumor and RT characteristics were stratified into groups. Patient age was organized into four groups (≤50, 51–60, 61–70 and >70 years of age). Comorbidity information was organized into three groups (0–1, 2 and ≥3 comorbidities) using the CDCS. Patient location of treatment was divided by US Census region. Population density of patient’s residence was categorized as metropolitan, urban and rural. Patient distance from reporting facility was stratified into four groups (<10, 10–19, 20–29 and ≥30 miles). Patient county income and educational levels were categorized as quartiles based on equally proportioned ranges among all US zip codes. Education was defined as the percentage of adults in the patient’s zip code without a high school diploma. When comparing fraction sizes, RT dose was dichotomized into 2 Gy fractions and 2.25 Gy fractions with all other fraction sizes being excluded.

### 4.4. Statistical Analysis

Variables associated with the receipt of IMRT or 3DCRT were assessed using nominal logistic regression. Overall survival (OS) was estimated using Kaplan–Meier methods and Cox proportional hazard models in the entire population and in a propensity matched cohort. Categorical variables were compared using chi-square tests. Continuous variables were compared using a two-sample *t*-test. Univariable and multivariable logistic regression were used to determine factors associated with the receipt of IMRT. Variables were included in the multivariable analysis if they were found to be associated with RT modality (*p* < 0.1) on univariable analysis. Overall survival (OS) was defined as the time from diagnosis to death. The Kaplan–Meier method and log-rank test were used to estimate and compare OS. Univariable and multivariable Cox proportional hazards modeling was used to determine factors associated with OS. Variables were included in the multivariable analysis if they were found to be associated with OS (*p* < 0.1) on univariable analysis. Sensitivity analysis using 1-to-1 propensity score matching was performed to ensure patients who received IMRT were equivalent to those who received 3DCRT with regard to all covariates. Groups were matched based on variables found to be statistically significant predictors of receipt of IMRT on univariate analysis. Analyses were performed using STATA MP 14 (StataCorp LLC, College Station, TX, USA) and SPSS (IBM Corporation, Armonk, NY, USA) statistical software. A *p* value less than 0.05 was considered statistically significant.

## 5. Conclusions

Although IMRT was associated with conventional fractionation, we observed similar survival between IMRT and 3DCRT, supporting the implementation of this potentially less toxic modality without compromising survival. A prospective comparison of IMRT and 3DCRT for patients with early stage glottic SCC is warranted.

## Figures and Tables

**Figure 1 cancers-11-01996-f001:**
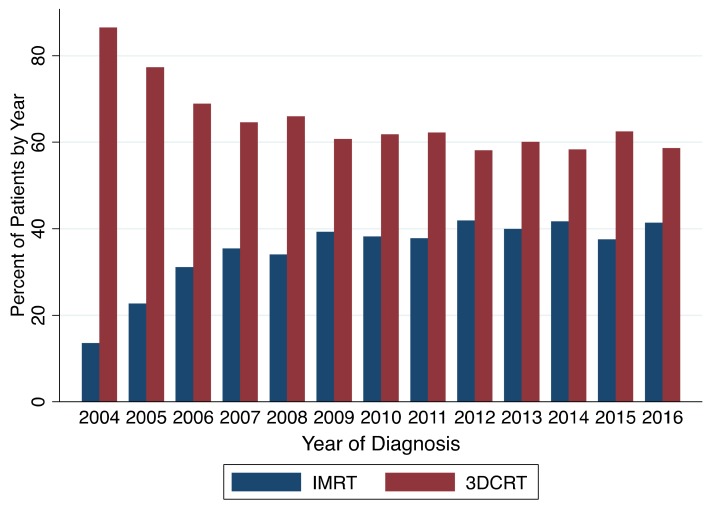
Utilization of IMRT and 3DCRT by year of diagnosis.

**Figure 2 cancers-11-01996-f002:**
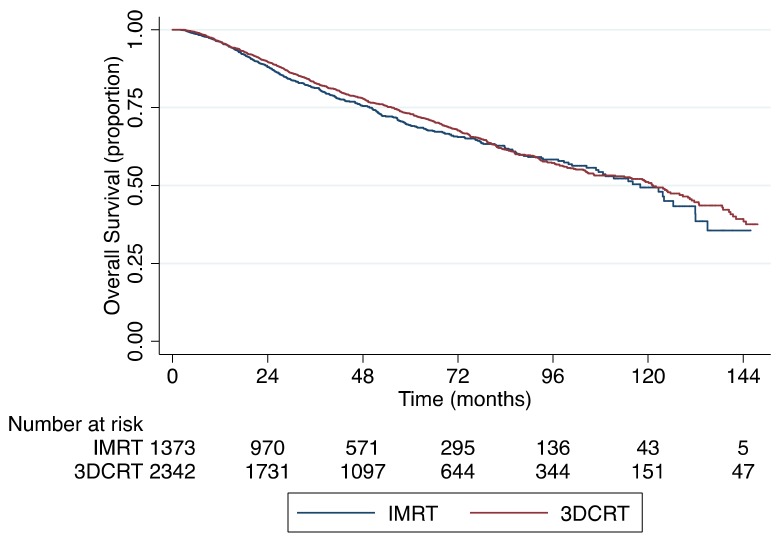
Kaplan–Meier overall survival curve stratified by IMRT vs. 3DCRT (*p* = 0.251).

**Figure 3 cancers-11-01996-f003:**
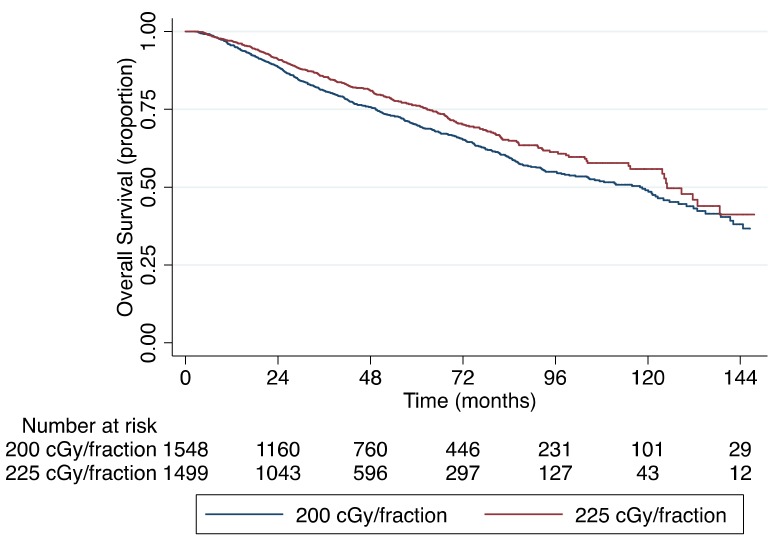
Kaplan–Meier overall survival curve stratified by fractionation size (*p* = 0.002).

**Figure 4 cancers-11-01996-f004:**
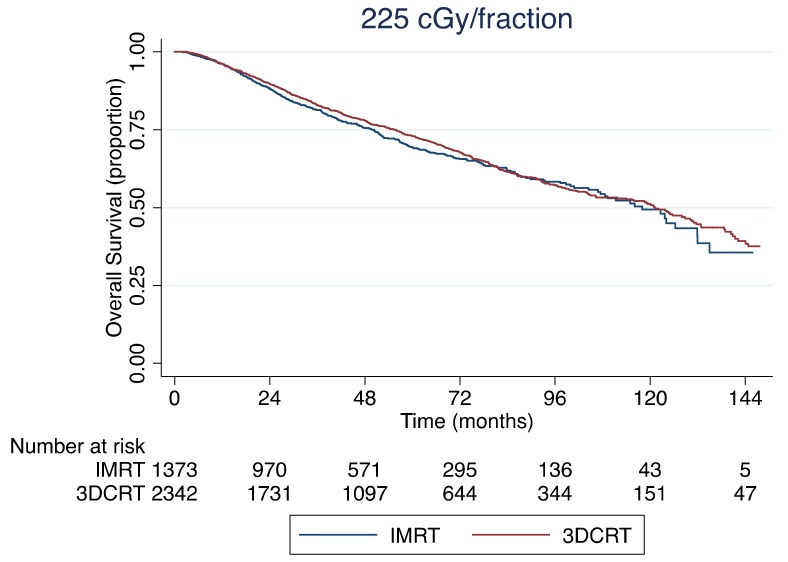
Kaplan–Meier overall survival curve stratified by IMRT vs. 3DCRT for only patients receiving 225 cGy per fraction (*p* = 0.342).

**Figure 5 cancers-11-01996-f005:**
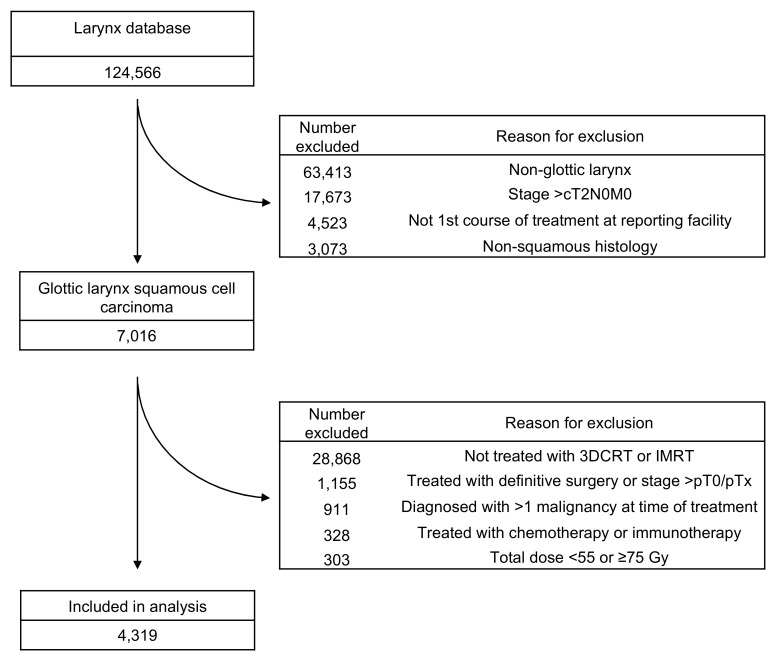
Consolidated Standard of Reporting Trials (CONSORT) diagram. 3DCRT, 3D conformal radiotherapy; IMRT, intensity-modulated radiotherapy.

**Table 1 cancers-11-01996-t001:** Baseline demographics and clinical characteristics by radiotherapy modality.

Characteristics	IMRT	3DCRT	*p* Value
	(*n* = 1623)	(*n* = 2696)	
Median follow-up (months)	39	44	0.0001
	(21–66)	(23–76)	
Age			0.009
≤50 y	148 (9.1%)	203 (7.5%)	
51–60 y	402 (24.8%)	675 (25.0%)	
61–70 y	564 (34.7%)	856 (31.8%)	
>70 y	509 (31.4%)	962 (35.7%)	
Gender			0.437
Male	1376 (84.8%)	2309 (85.7%)	
Female	247 (15.2%)	387 (14.3%)	
Race			0.004
White	1360 (83.8%)	2355 (87.3%)	
Black	209 (12.9%)	264 (9.8%)	
Other	54 (3.3%)	77 (2.9%)	
Ethnicity			0.001
Non-Hispanic	1449 (89.3%)	2486 (92.2%)	
Hispanic	94 (5.8%)	92 (3.4%)	
Unknown	80 (4.9%)	118 (4.4%)	
Insurance			0.108
Uninsured	61 (3.8%)	80 (3.0%)	
Private	620 (38.2%)	1008 (37.4%)	
Medicaid	90 (5.5%)	130 (4.8%)	
Medicare	766 (47.2%)	1362 (50.5%)	
Other/Unknown	86 (5.3%)	116 (4.3%)	
Median income			0.179
<$38 k	331 (20.4%)	492 (18.3%)	
$38–47.9 k	391 (24.1%)	721 (26.7%)	
$48–62.9 k	425 (26.2%)	721 (26.7%)	
≥$63 k	468 (28.8%)	753 (27.9%)	
Unknown	8 (0.5%)	9 (0.4%)	
Education ^a^			0.126
≥21%	298 (18.4%)	426 (15.8%)	
13–20.9%	459 (28.3%)	735 (27.3%)	
7–12.9%	541 (33.3%)	945 (35.0%)	
<7%	319 (19.6 %)	582 (21.6%)	
Unknown	6 (0.4%)	8 (0.3%)	
Location			0.032
South	594 (36.6%)	886 (32.9%)	
Northeast	394 (24.3%)	635 (23.5%)	
Midwest	454 (30.0%)	820 (30.4%)	
West	163 (10.0%)	328 (12.2%)	
Unknown	18 (1.1%)	27 (1.0%)	
Population			0.517
Metropolitan	1312 (80.8%)	2164 (80.3%)	
Urban	253 (15.6%)	453 (16.8%)	
Rural	30 (1.9%)	41 (1.5%)	
Unknown	28 (1.7%)	38 (1.4%)	
Year of diagnosis			<0.0001
2004–2007	189 (11.7%)	495 (18.4%)	
2008–2011	452 (27.8%)	756 (28.0%)	
2012–2016	982 (60.5%)	1445 (53.6%)	
Distance			0.209
<10 m	859 (52.9%)	1505 (55.8%)	
10–19 m	362 (22.3%)	575 (21.3%)	
20–29 m	167 (10.3%)	279 (10.4%)	
≥30 m	229 (14.1%)	332 (12.3%)	
Unknown	6 (0.4%)	5 (0.2%)	
Facility			<0.0001
Community	141 (8.7%)	202 (7.5%)	
Comprehensive	689 (42.4%)	1393 (51.7%)	
Academic	532 (32.8%)	644 (23.9%)	
Integrated	243 (15.0%)	430 (15.9%)	
Other	18 (1.1%)	27 (1.0%)	
CDCS			0.961
0–1	1519 (93.6%)	2528 (93.8%)	
2	74 (4.6%)	118 (4.4%)	
≥3	30 (1.8%)	50 (1.8%)	
Clinical T-stage			<0.0001
cT0	64 (3.9%)	180 (6.7%)	
cT1–2	1559 (96.1%)	2516 (93.3%)	
Fraction size			<0.0001
≤2 Gy	816 (50.3%)	1177 (43.7%)	
>2 Gy	807 (49.7%)	1519 (56.3%)	
Fraction size			<0.0001
2 Gy	691 (42.6%)	1032 (38.3%)	
2.25 Gy	576 (35.5%)	1279 (47.4%)	
Other	356 (21.9%)	385 (14.3%)	
Median total dose	66 (IQR, 63–70)	65 (IQR, 63–66)	<0.0001

^a^ Percentage of adults in the patient’s zip code without a high school diploma. Abbreviations: IMRT, intensity-modulated radiotherapy; 3DCRT, 3D conformal radiotherapy; CDCS, Charlson-Deyo comorbidity score; y, year; Gy, Gray; IQR, interquartile range.

**Table 2 cancers-11-01996-t002:** Univariable and multivariable logistic regression for factors associated with receipt of IMRT (*n* = 4319).

Covariate	Univariate Likelihood of Receiving IMRT		Multivariate Likelihood of Receiving IMRT	
	Odds Ratio	*p* Value	Odds Ratio	*p* Value
Age				
≤50 y	Reference		Reference	
51–60 y	0.82 (0.64–1.04)	0.106	0.87 (0.64–1.17)	0.341
61–70 y	0.90 (0.71–1.15)	0.403	1.00 (0.74–1.36)	0.988
>70 y	0.72 (0.59–0.90)	0.008	0.82 (0.60–1.14)	0.246
Gender				
Male	Reference		N.D.	
Female	1.07 (0.90–1.27)	0.437		
Race				
White	Reference		Reference	
Black	1.37 (1.12–1.66)	0.001	1.32 (1.04–1.67)	0.023
Other	1.21 (0.85–1.73)	0.283	0.94 (0.62–1.42)	0.769
Ethnicity				
Non-Hispanic	Reference		Reference	
Hispanic	1.75 (1.31–2.35)	<0.0001	1.77 (1.24–2.52)	0.002
Unknown	1.16 (0.87–1.56)	0.309	1.38 (0.98–1.95)	0.067
Insurance				
Uninsured	Reference		Reference	
Private	0.81 (0.57–1.14)	0.226	0.92 (0.60–1.42)	0.717
Medicaid	0.91 (0.59–1.39)	0.658	0.78 (0.46–1.32)	0.358
Medicare	0.74 (0.52–1.04)	0.084	0.88 (0.57–1.38)	0.585
Other/Unknown	0.97 (0.63–1.50)	0.899	1.03 (0.61–1.75)	0.901
Median income				
<$38 k	Reference		Reference	
$38–47.9 k	0.81 (0.67–0.97)	0.023	0.85 (0.67–1.08)	0.176
$48–62.9 k	0.88 (0.73–1.05)	0.159	1.03 (0.80–1.33)	0.798
≥$63 k	0.92 (0.77–1.11)	0.391	1.24 (0.93–1.67)	0.134
Education ^a^				
≥21%	Reference		Reference	
13–20.9%	0.89 (0.74–1.08)	0.238	1.02 (0.80–1.29)	0.889
7–12.9%	0.82 (0.68–0.98)	0.031	0.90 (0.69–1.17)	0.429
<7%	0.78 (0.64–0.96)	0.018	0.79 (0.58–1.08)	0.144
Location				
South	Reference		Reference	
Northeast	0.93 (0.79–1.09)	0.352	1.12 (0.92–1.37)	0.26
Midwest	0.83 (0.71–0.96)	0.015	0.98 (0.81–1.18)	0.833
West	0.74 (0.60–0.92)	0.006	0.86 (0.66–1.11)	0.237
Population				
Metropolitan	Reference		N.D.	
Urban	0.92 (0.78–1.09)	0.339		
Rural	1.21 (0.75–1.94)	0.439		
Year of diagnosis				
2004–2007	Reference		Reference	
2008–2011	1.57 (1.28–1.92)	<0.0001	2.00 (1.54–2.60)	<0.0001
2012–2016	1.77 (1.48–2.14)	<0.0001	2.77 (2.17–3.55)	<0.0001
Distance				
<10 m	Reference		Reference	
10–19 m	1.10 (0.94–1.29)	0.218	1.14 (0.95–1.37)	0.155
20–29 m	1.05 (0.85–1.29)	0.656	1.14 (0.90–1.45)	0.281
≥30 m	1.21 (1.00–1.46)	0.048	1.15 (0.91–1.45)	0.246
Facility				
Community	Reference		Reference	
Comprehensive	0.71 (0.56–0.90)	0.004	0.72 (0.55–0.95)	0.019
Academic	1.18 (0.93–1.51)	0.176	1.08 (0.81–1.43)	0.62
Integrated	0.81 (0.62–1.06)	0.12	0.83 (0.61–1.13)	0.229
CDCS				
0–1	Reference		N.D.	
2	1.04 (0.78–1.41)	0.778		
≥3	1.00 (0.63–1.58)	0.995		
Clinical T-stage				
cT0	Reference		Reference	
cT1–2	1.74 (1.30–2.33)	<0.0001	0.56 (0.12–2.57)	0.451
Fraction size				
2 Gy	Reference		Reference	
2.25 Gy	0.77 (0.68–0.87)	<0.0001	0.56 (0.48–0.65)	<0.0001

^a^ Percentage of adults in the patient’s zip code without a high school diploma. Abbreviations: IMRT, intensity-modulated radiotherapy; 3DCRT, 3D conformal radiotherapy; CDCS, Charlson-Deyo comorbidity score; y, year; Gy, Gray; m, miles; N.D., not determined.

**Table 3 cancers-11-01996-t003:** Univariable and multivariable analysis for factors associated with survival (*n* = 4319).

Covariate	Univariate Analysis for Survival		Multivariate Analysis for Survival	
	Hazard Ratio	*p* Value	Hazard Ratio	*p* Value
Age				
≤50 y	Reference		Reference	
51–60 y	1.20 (0.87–1.64)	0.263	1.11 (0.77–1.61)	0.575
61–70 y	1.38 (1.02–1.87)	0.036	1.05 (0.73–1.52)	0.787
>70 y	3.23 (2.42–4.32)	<0.0001	2.17 (1.49–3.15)	<0.0001
Gender				
Male	Reference		Reference	
Female	0.83 (0.69–0.99)	0.04	0.84 (0.68–1.03)	0.099
Race				
White	Reference		Reference	
Black	1.00 (0.81–1.22)	0.976	0.90 (0.71–1.15)	0.393
Other	0.65 (0.41–1.02)	0.06	0.63 (0.39–1.03)	0.068
Ethnicity				
Non-Hispanic	Reference		N.D.	
Hispanic	0.78 (0.55–1.10)	0.154		
Unknown	0.88 (0.67–1.16)	0.366		
Insurance				
Uninsured	Reference		Reference	
Private	0.73 (0.48–1.12)	0.149	0.60 (0.37–1.00)	0.049
Medicaid	1.60 (0.98–2.62)	0.06	1.09 (0.61–1.95)	0.772
Medicare	1.86 (1.22–2.81)	0.004	1.06 (0.64–1.76)	0.813
Other/Unknown	1.84 (1.13–3.00)	0.015	1.51 (0.85–2.67)	0.156
Median income				
<$38 k	Reference		Reference	
$38–47.9 k	0.89 (0.74–1.06)	0.192	0.79 (0.64–0.98)	0.034
$48–62.9 k	0.84 (0.70–1.01)	0.059	0.77 (0.62–0.96)	0.02
≥$63 k	0.76 (0.64–0.91)	0.003	0.78 (0.62–0.97)	0.028
Education ^a^				
≥21%	Reference		N.D.	
13–20.9%	0.97 (0.80–1.17)	0.745		
7–12.9%	0.96 (0.80–1.15)	0.64		
<7%	0.93 (0.76–1.13)	0.456		
Location				
South	Reference		N.D.	
Northeast	0.99 (0.84–1.17)	0.934		
Midwest	1.09 (0.94–1.27)	0.244		
West	1.16 (0.95–1.43)	0.15		
Population				
Metropolitan	Reference		Reference	
Urban	1.19 (1.02–1.39)	0.029	0.97 (0.79–1.18)	0.752
Rural	0.95 (0.57–1.58)	0.831	0.86 (0.47–1.58)	0.633
Year of diagnosis				
2004–2007	Reference		N.D.	
2008–2011	1.01 (0.87–1.18)	0.904		
2012–2016	1.05 (0.88–1.26)	0.562		
Distance				
<10 m	Reference		N.D.	
10–19 m	0.92 (0.78–1.08)	0.293		
20–29 m	1.03 (0.84–1.26)	0.784		
≥30 m	1.07 (0.89–1.29)	0.463		
Facility				
Community	Reference		Reference	
Comprehensive	0.92 (0.74–1.16)	0.487	1.11 (0.85–1.44)	0.455
Academic	0.71 (0.55–0.91)	0.007	0.88 (0.65–1.19)	0.403
Integrated	0.73 (0.56–0.95)	0.018	0.85 (0.62–1.16)	0.291
CDCS				
0–1	Reference		Reference	
2	2.36 (1.85–3.00)	<0.0001	1.76 (1.35–2.29)	<0.0001
≥3	2.04 (1.28–3.25	0.003	1.73 (0.99–3.02)	0.052
Clinical T-stage				
cT0	Reference		Reference	
cT1–2	1.36 (1.01–1.82)	0.041	0.46 (0.10–2.09)	0.313
Fraction size				
2 Gy	Reference		Reference	
2.25 Gy	0.83 (0.73–0.94)	0.003	0.78 (0.69–0.92)	0.003
Modality				
3DCRT	Reference		Reference	
IMRT	1.08 (0.95–1.22)	0.251	1.08 (0.93–1.26)	0.302

^a^ Percentage of adults in the patient’s zip code without a high school diploma. Abbreviations: IMRT, intensity-modulated radiotherapy; 3DCRT, 3D conformal radiotherapy; CDCS, Charlson-Deyo comorbidity score; y, year; Gy, Gray; m, miles; N.D., not determined.

**Table 4 cancers-11-01996-t004:** Propensity score matched patient characteristics (*n* = 1428).

Covariate	IMRT	3DCRT	*p* Value
	(*n* = 714)	(*n* = 714)	
Median follow-up (months)	35 (27–46)	33 (25–42)	0.983
Age			0.914
≤50 y	29 (4.1%)	24 (3.4%)	
51–60 y	178 (24.9%)	179 (25.1%)	
61–70 y	241 (33.7%)	240 (33.6%)	
>70 y	266 (37.3%)	271 (38.0%)	
Race			0.83
White	659 (92.3%)	660 (92.4%)	
Black	54 (7.6%)	52 (7.3%)	
Other	1 (0.1%)	2 (0.3%)	
Ethnicity			1
Non-Hispanic	704 (98.6%)	704 (98.6%)	
Hispanic	10 (1.4%)	10 (1.4%)	
Location			0.993
South	261 (36.6%)	258 (36.1%)	
Northeast	172 (24.1%)	170 (23.8%)	
Midwest	231 (32.3%)	234 (32.8%)	
West	50 (7.0%)	52 (7.3%)	
Year of diagnosis			0.992
2004–2007	56 (7.8%)	57 (8.0%)	
2008–2011	189 (26.5%)	190 (26.6%)	
2012–2016	469 (65.7%)	467 (65.4%)	
Facility			0.996
Community	43 (6.0%)	43 (6.0%)	
Comprehensive	374 (52.4%)	373 (52.3%)	
Academic	212 (29.7%)	210 (29.4%)	
Integrated	85 (11.9%)	88 (12.3%)	
Clinical T-stage			0.681
cT0	13 (1.8%)	11 (1.5%)	
cT1–2	701 (98.2%)	703 (98.5%)	
Fraction size			0.958
2 Gy	351 (49.2%)	352 (49.3%)	
2.25 Gy	363 (50.8%)	362 (50.7%)	

^a^ Percentage of adults in the patient’s zip code without a high school diploma. Abbreviations: IMRT, intensity-modulated radiotherapy; 3DCRT, 3D conformal radiotherapy; y, year; Gy, Gray.

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
