# Peer review of "Patterns of Care and Outcomes of Intensity-Modulated Radiotherapy and 3D Conformal Radiotherapy for Early Stage Glottic Cancer: A National Cancer Database Analysis"

_cancers, 2019, doi:10.3390/cancers11121996_

Round 1

Reviewer 1 Report

Cancers review - 628295

28 October 2019

Patterns of care and outcomes of intensity modulated radiotherapy and 3D conformal radiotherapy for early stage glottic cancer:  a national cancer database analysis

OVERALL

This study was a cohort comparison retrospective evaluation of patients treated for early stage glottic cancer across the United States, as entered in the National Cancer Database.  This demonstrates favourable treatment of early glottic cancer with IMRT, that is at least oncologically equivalent to traditional 3DCRT, and further indicates some demographic factors associated with receiving IMRT treatment (largely race, type of treating facility and geography).  An indication that a hypofractionated treatment schedule is more efficacious in eradicating early glottic cancer warrants further study and an RCT.

Thorough data analysis has been performed on data that can be derived from this database, and the paper is well written, with clear findings and appropriate statistical methodology.  The findings are relevant to all physicians treating laryngeal cancer and should inform future research directions, in particular, randomised study protocols.  Limitations are included and justified by the methodology being utilized.

Excellent paper.  I suggest publication as written.

ABSTRACT                

Excellent

INTRODUCTION
Appropriate and clear.  Only issue is that this database does not provide information about treatment toxicity that is clearly an important consideration in selection of treatment regimens.

METHODS

Appropriately described.

RESULTS

Appropriately described.  Includes CONSORT diagram.

DISCUSSION

Satisfactory.  Includes limitations and importantly, that this paper cannot estimate treatment toxicity, which is an important consideration in selection of therapy modality.

CONCLUSION

Appropriate based on current available data.

REFERENCES

Satisfactory.

FIGURES

Tables are satisfactory.

Author Response

We thank the Reviewer for their comments.

Reviewer 2 Report

This study rise the relevant question of the best modality of radiation therapy in the treatment of early-stage glottic cancer.

The authors compared IMRT vs. 3RCRT in terms of overall survival (OS) using the NCDB. They found no significant difference of OS between the 2 modalities of RT. They concluded that their results support the implementation of this potentially less toxic modality of RT in the treatment of patients with early-stage glottic carcinomas.

Overall, the methodology used to compare OS in the 2 group of treatment seems appropriate.

As stipulated by the authors, the goal of IMRT is to "reduce toxicity without compromising treatment outcomes". However, using the NCDB, the authors were not able to evaluate treatment-related toxicity and could only evaluate OS in terms of oncologic outcomes. I definitly believe that OS is not a relevant end-point to evaluate oncologic results of a RT technique in patients with early-stage glottic cancers. Treatment failure for these patients means local recurrence or total laryngectomy and not patient death. The appropriate end-points to evaluate in this type of study are: local recurrence, recurrence-free survival, larynx preservation rate... Using the NCDB, the authors will not be able to investigate these more relevant end-points.

To conclude, I believe that the use of the NCDB is not a relevant method to compare IMRT vs. 3DCRT in patients with early glottic cancers.

Author Response

This study rise the relevant question of the best modality of radiation therapy in the treatment of early-stage glottic cancer.

The authors compared IMRT vs. 3RCRT in terms of overall survival (OS) using the NCDB. They found no significant difference of OS between the 2 modalities of RT. They concluded that their results support the implementation of this potentially less toxic modality of RT in the treatment of patients with early-stage glottic carcinomas.

Overall, the methodology used to compare OS in the 2 group of treatment seems appropriate.

As stipulated by the authors, the goal of IMRT is to "reduce toxicity without compromising treatment outcomes". However, using the NCDB, the authors were not able to evaluate treatment-related toxicity and could only evaluate OS in terms of oncologic outcomes. I definitly believe that OS is not a relevant end-point to evaluate oncologic results of a RT technique in patients with early-stage glottic cancers. Treatment failure for these patients means local recurrence or total laryngectomy and not patient death. The appropriate end-points to evaluate in this type of study are: local recurrence, recurrence-free survival, larynx preservation rate... Using the NCDB, the authors will not be able to investigate these more relevant end-points.

To conclude, I believe that the use of the NCDB is not a relevant method to compare IMRT vs. 3DCRT in patients with early glottic cancers.

We thank the Reviewer for their comments. This manuscript presents two major points for IMRT in early stage glottic cancer: (1) implementation and practice patterns and (2) survival outcomes, or lack thereof, associated with this technique. Furthermore, the use of the NCDB dataset in this manuscript is relevant because IMRT for early stage glottic cancers is an under studied area as evidenced by only 22 publications currently in PubMed.gov that address this topic (https://www.ncbi.nlm.nih.gov/pubmed/?term=intensity+modulated++radiotherapy++glottic+cancer++early+stage). Furthermore, few reports address practice patterns and/or outcomes as many of these reports are either case studies or focus on dosimentric techniques. Given that this is an understudied topic, we maintain that the NCDB is a relevant method to compare IMRT with more standard techniques such as 3DCRT in looking at practice patterns as well as survival, even if survival is a less informative endpoint to assess outcomes.

                We agree with the reviewer that survival is a less informative endpoint to evaluate the oncologic results of radiotherapy techniques in patients with early-stage glottic cancers. We agree that patient data for toxicity, local control, and cancer-specific survival would better facilitate comparisons between IMRT and 3DCRT. Nevertheless, disease recurrence in early stage glottic cancers does impact survival [1], even if this impact is small. Consequently, the large number of patients in the NCDB facilitates the detection of small differences in survival associated with different radiotherapy techniques. To this end, using the NCDB, we and others observed that patients treated with 200 cGy fraction sizes had worse survival differences than patients treated with 225 cGy fraction sizes (i.e. different radiotherapy techniques). These survival differences were not observed in the original randomized trials addressing this issue [2–5].   We state this in lines 157-167.

References

Ganly, I.; Patel, S.G.; Matsuo, J.; Singh, B.; Kraus, D.H.; Boyle, J.O.; Wong, R.J.; Shaha, A.R.; Lee, N.; Shah, J.P. Results of surgical salvage after failure of definitive radiation therapy for early-stage squamous cell carcinoma of the glottic larynx. Arch. Otolaryngol. Head. Neck Surg. 2006, 132, 59–66. Yamazaki, H.; Nishiyama, K.; Tanaka, E.; Koizumi, M.; Chatani, M. Radiotherapy for early glottic carcinoma (T1N0M0): Results of prospective randomized study of radiation fraction size and overall treatment time. Int. J. Radiat. Oncol. 2006, 64, 77–82. Bledsoe, T.J.; Park, H.S.; Stahl, J.M.; Yarbrough, W.G.; Burtness, B.A.; Decker, R.H.; Husain, Z.A. Hypofractionated Radiotherapy for Patients with Early-Stage Glottic Cancer: Patterns of Care and Survival. JNCI J. Natl. Cancer Inst. 2017, 109. Stokes, W.A.; Stumpf, P.K.; Jones, B.L.; Blatchford, P.J.; Karam, S.D.; Lanning, R.M.; Raben, D. Patterns of fractionation for patients with T2N0M0 glottic larynx cancer undergoing definitive radiotherapy in the United States. Oral Oncol. 2017, 72, 110–116. Stokes, W.A.; Abbott, D.; Phan, A.; Raben, D.; Lanning, R.M.; Karam, S.D. Patterns of Care for Patients With Early-Stage Glottic Cancer Undergoing Definitive Radiation Therapy: A National Cancer Database Analysis. Int. J. Radiat. Oncol. Biol. Phys. 2017, 98, 1014–1021.

Reviewer 3 Report

The authors describe a United States national cancer database analysis of IMRT vs 3DCRT for glottic gancer in early stages. The subject is of interest for Radiation Oncologists and ENT specialists and sheds some light into the intramodal comparison.

Some minor points:

a database analysis lacks toxicity data, which is a weekness of the study I would rather call 2.25 Gy a slight hypofractionation (please provide a range in M&M). Was 2.25 Gy on the SIB volume? What was the average baseplan dose? Since alterations in fractionation are a major conclusion of the manuscript more information should be provided.  no information of previous local treatments are provided. It is likely that a sign. subset received previous laser surgical resection.

Author Response

The authors describe a United States national cancer database analysis of IMRT vs 3DCRT for glottic gancer in early stages. The subject is of interest for Radiation Oncologists and ENT specialists and sheds some light into the intramodal comparison.

Some minor points:

a database analysis lacks toxicity data, which is a weekness of the study

We thank the Reviewer for their comments. We address this limitation in the Discussion (lines 159-161).

I would rather call 2.25 Gy a slight hypofractionation (please provide a range in M&M).

We thank the Reviewer for their comments. We now call 2.25 Gy a slightly hypofractionated regimen throughout the manuscript.

Was 2.25 Gy on the SIB volume? What was the average baseplan dose?

We thank the Reviewer for their comments. The NCDB provides only the regional and boost radiation doses. Consequently, there is no information on SIB volumes. We now include the median total dose for patients treated with IMRT and 3DCRT in Table 1 as well as for patients treated with 2.25 Gy or 2 Gy fraction sizes in lines 57-59. 

Since alterations in fractionation are a major conclusion of the manuscript more information should be provided. 

We thank the Reviewer for their comments. We now describe the range in fraction sizes and median fraction sizes for patients treated with IMRT or 3DCRT. We state this in lines 55-57. Furthermore, for the comparison of different fraction sizes (Figures 3 & 4), we only included the two most commonly used fractionation regimens: 2 Gy or 2.25 Gy fractions. We state this in lines 196-197. 

No information of previous local treatments are provided. It is likely that a sign. subset received previous laser

surgical resection.

We thank the Reviewer for their comments. We set several inclusion and exclusion criteria to avoid including patients who had received previous treatment. First, we included only patients who received their first course of treatment at the reporting facility. Second, we excluded patients receiving any surgery to avoid patients undergoing any invasive procedures other than a biopsy. Finally, we excluded patients with pTis, pT1 or pT2 which would suggest they had undergone a surgical procedure. These points are demonstrated in Figure 5 and in lines 179-181.
